# Efficient Learning of Discrete-Continuous Computation Graphs

**David Friede[1,2]**
david@informatik.uni-mannheim.de

**[2]University of Mannheim**
Mannheim, Germany

**Mathias Niepert[1]**
mathias.niepert@neclab.eu

**[1]NEC Laboratories Europe**
Heidelberg, Germany

## Abstract

Numerous models for supervised and reinforcement learning benefit from combinations of discrete and continuous model components. End-to-end learnable discrete-continuous models are compositional, tend to generalize better, and are more interpretable. A popular approach to building discrete-continuous computation graphs is that of integrating discrete probability distributions into neural networks using stochastic softmax tricks. Prior work has mainly focused on computation graphs with a single discrete component on each of the graph's execution paths. We analyze the behavior of more complex stochastic computations graphs with multiple sequential discrete components. We show that it is challenging to optimize the parameters of these models, mainly due to small gradients and local minima. We then propose two new strategies to overcome these challenges. First, we show that increasing the scale parameter of the Gumbel noise perturbations during training improves the learning behavior. Second, we propose dropout residual connections specifically tailored to stochastic, discrete-continuous computation graphs. With an extensive set of experiments, we show that we can train complex discrete-continuous models which one cannot train with standard stochastic softmax tricks. We also show that complex discrete-stochastic models generalize better than their continuous counterparts on several benchmark datasets.

## 1 Introduction

Neuro-symbolic learning systems aim to combine discrete and continuous operations. The majority of recent research has focused on integrating neural network components into probabilistic logics [28, 20], that is, making logic-based reasoning approaches more amenable to noisy and high-dimensional input data. On the other end of the spectrum are the data-driven approaches, which aim to learn a modular and discrete program structure in an end-to-end neural network based system. The broader vision followed by proponents of these approaches are systems capable of assembling the required modular operations to solve a variety of tasks with heterogeneous input data. Reinforcement learning [33], neuro-symbolic program synthesis [25], and neural module networks [1] are instances of such data-integrated discrete-continuous learning systems.

We focus on learning systems comprised of both symbolic and continuous operations where the symbolic operations are modeled as *discrete* probability distributions. The resulting systems can be described by their *stochastic computation graphs*, a formalism recently introduced to unify several related approaches [30]. Figure 1 illustrates two instances of such stochastic computation graphs. For a given high-dimensional input such as a set of images or a list of symbols, a computation graph, consisting of stochastic (discrete) and continuous (neural) nodes, is created to solve a particular task. Both, the mechanism to assemble the graphs (if not already provided) and the various operations are

35th Conference on Neural Information Processing Systems (NeurIPS 2021).

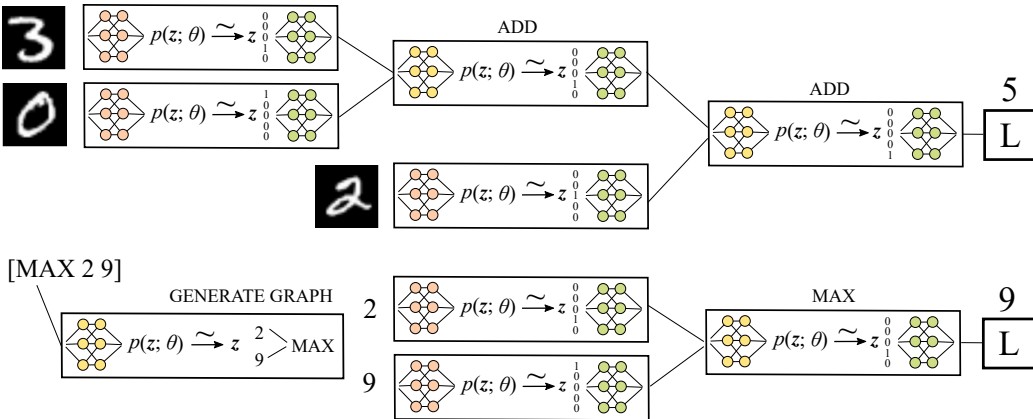

Figure 1: Two instances of discrete-continuous computation graphs. Top: The MNIST addition task aims to learn a neuro-symbolic program end-to-end that sums two or more digits only based on raw image inputs. The input images are mapped to the parameters of a categorical probability distribution (orange neural network). Samples from said distribution are mapped to learned vector representations of digits (green neural network). The addition module ADD takes the vector representations of two digits and learns to compute the sum of the corresponding digits. Bottom: The ListOps task aims to learn the solution to an arithmetic expression. Given the syntactic input and the numerical solution, we learn the latent parse tree, which consists of discrete and continuous components modeling the operations MAX, MIN, and MED. More details are provided in the experimental section.

learned end-to-end. Discrete nodes in the computation graph are Gibbs distributions modeled at a particular temperature which can also be used to model the argmax operation at zero temperature.

The majority of prior work has focused on graphs with a single discrete probability distribution along each execution path. Examples are the discrete variational autoencoder [14], learning to explain [4], and other applications of stochastic softmax tricks [26]. We aim to analyze and improve the training behavior of *complex* computation graphs, that is, graphs with more than one discrete probability distribution in its execution paths. More concretely, we focus on computation graphs where the stochastic nodes are categorical random variables modeled with the Gumbel-softmax trick [14, 19]. In Section 2, we show both analytically and empirically that it is challenging to optimize the parameters of these models, mainly due to insufficient gradient signals often caused by local minima and saturation. We propose two new methods for mitigating the causes of poor optimization behavior. First, we show in Section 2.1 that increasing the scale parameter $\beta$ of the Gumbel noise perturbations improves the models' learning behavior. Increasing $\beta$ increases the probability of escaping local minima during training. Second, we propose dropout residual connections for discrete-continuous computation graphs in Section 2.2. By randomly skipping some discrete distributions, we provide more informative gradients throughout the full computation graph. We show empirically for several complex discrete-continuous models that the proposed methods are required for training. We also show that the resulting discrete-continuous models generalize better and significantly outperform state of the art approaches on several benchmark datasets.

## 2 Efficient Learning of Discrete-Continuous Computation Graphs

Standard neural networks compose basic differentiable functions. These networks, therefore, can be fully described by a directed acyclic graph (the computation graph) that determines the operations executed during the forward and backward passes. Schulman et al. [30] proposed *stochastic computation graphs* as an extension of neural networks that combine deterministic and stochastic operations – a node in the computation graph can be either a differentiable function or a probability distribution. A *stochastic node* $X$ is typically a random variable with a parameterized probability distribution $p_{\boldsymbol{\theta}(x)}$ with parameters $\boldsymbol{\theta}$. Suppose that $f$ is a smooth function (such as the loss function of a learning problem), then the gradient of $f$ at the stochastic node is $\nabla_{\boldsymbol{\theta}} \mathbb{E}_{p_{\boldsymbol{\theta}}(x)}[f(x)]$. Kingma and Welling [15] proposed the *reparameterization trick* to overcome the problem of computing gradients with respect to the parameters of a distribution. The idea is to find a function $g$ and distribution $\rho$ such that one

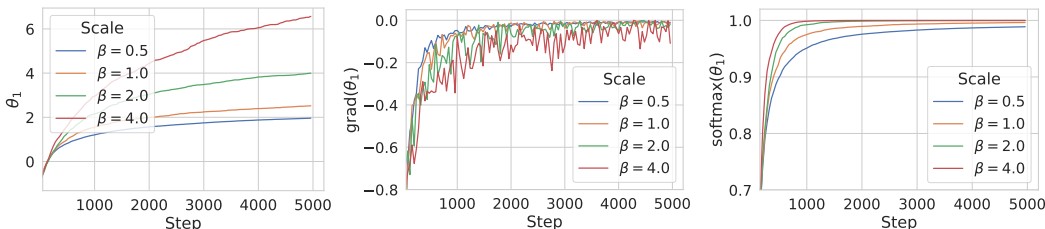

Figure 2: The values of the parameters, their gradients, and the softmax probabilities for various values of $\beta$ (the scale parameter) for a constant Gumbel-Softmax parameter $\tau$. Early during training, a lower scale parameter $\beta$ (relative to $\tau$) works well and has less variance. Once the probabilities saturate, however, we can only continue to obtain a sufficient gradient signal for larger values of $\beta$.

can replace $x \sim p_{\boldsymbol{\theta}}(x)$ by $x = g(z, \boldsymbol{\theta})$ with $z \sim \rho(z)$. If this is possible, then one can write

$$\nabla_{\boldsymbol{\theta}} \mathbb{E}_{x \sim p_{\boldsymbol{\theta}}(x)}[f(x)] = \mathbb{E}_{z \sim \rho(z)}[\nabla_{\boldsymbol{\theta}} f(g(z, \boldsymbol{\theta}))] \approx \frac{1}{S} \sum_{i=1}^{S} \nabla_{\boldsymbol{\theta}} f(g(z_i, \boldsymbol{\theta})) \text{ with } z_i \sim \rho(z). \quad (1)$$

With this paper we address the problem of training stochastic computation graphs where the stochastic nodes are based on categorical (discrete) distributions approximated using stochastic softmax tricks [14, 26]. More specifically, we consider discrete-continuous components modeling a categorical variable $X$ with $k$ possible values $\boldsymbol{z}_1, ..., \boldsymbol{z}_k$. Each of the $\boldsymbol{z}_i$ is the one-hot encoding of the category $i$. We consider discrete-continuous functions[1] $f : \mathbb{R}^n \to \mathbb{R}^n$ with $\boldsymbol{v} = f(\boldsymbol{u})$ defined as follows

$$\boldsymbol{\theta} = g_{\boldsymbol{w}}(\boldsymbol{u}) \quad (2a)$$

$$p(\boldsymbol{z}_i; \boldsymbol{\theta}) = \frac{\exp(\boldsymbol{\theta}_i)}{\sum_{j=1}^{k} \exp(\boldsymbol{\theta}_j)} \quad (2b)$$

$$\boldsymbol{z} \sim p(\boldsymbol{z}; \boldsymbol{\theta}) \quad (2c)$$

$$\boldsymbol{v} = h_{\boldsymbol{w}'}(\boldsymbol{z}) \quad (2d)$$

$\boldsymbol{u}$ ⟶ $\theta$ ⟶ $p(z; \theta)$ ⟶ $\boldsymbol{z}$ ⟶ $\boldsymbol{v}$
$g$      $h$
$f$

Figure 3: Illustration of generic discrete-continuous component (function $f$).

We assume that the functions $g$ and $h$ (parameterized by $\boldsymbol{w}$ and $\boldsymbol{w}'$) are expressed using differentiable neural network components. In the majority of cases, $h$ is defined as $\boldsymbol{z}^{\mathsf{T}} \boldsymbol{w}'$ for a learnable matrix $\boldsymbol{w}'$, mapping object $\boldsymbol{z}_i$ to its learned vector representation. Figure 3 illustrates a generic discrete-continuous component.

Let $\mathrm{Gumbel}(0, \beta)$ be the Gumbel distribution with location $0$ and scale $\beta$. Using the Gumbel-max trick, we can sample $\boldsymbol{z} \sim p(\boldsymbol{z}; \boldsymbol{\theta})$ as follows:

$$\boldsymbol{z} := \boldsymbol{z}_i \quad \text{with} \quad i = \arg\max_{j \in \{1, ..., k\}} [\boldsymbol{\theta}_j + \boldsymbol{\epsilon_j}] \text{ where } \boldsymbol{\epsilon_j} \sim \mathrm{Gumbel}(0, 1). \quad (3)$$

The Gumbel-softmax trick is a relaxation of the Gumbel-max trick (relaxing the argmax into a softmax with a scaling parameter $\lambda$) that allows one to perform standard backpropagation:

$$\boldsymbol{z}_i = \frac{\exp\left((\boldsymbol{\theta}_i + \boldsymbol{\epsilon_i})/\tau\right)}{\sum_{j=1}^{k} \exp\left((\boldsymbol{\theta}_j + \boldsymbol{\epsilon_j})/\tau\right)} \text{ where } \boldsymbol{\epsilon_i} \sim \mathrm{Gumbel}(0, 1). \quad (4)$$

Hence, instead of sampling a discrete $\boldsymbol{z}$, the Gumbel-Softmax trick computes a relaxed $\boldsymbol{z}$ in its place.

We are concerned with the analysis of the behavior of the Gumbel-softmax trick in more complex stochastic computation graphs. In these computation graphs, multiple sequential Gumbel-softmax components occur on a single execution path. Throughout the remainder of the paper, we assume that during training, we use the Softmax-trick as outlined above, while at test time, we sample discretely using the Gumbel-max trick. Depending on the use case, we also sometimes compute the argmax at test time instead of samples from the distribution.

Let us first take a look at $\partial \boldsymbol{v} / \partial \boldsymbol{u}$, that is, the partial derivative of the output of the discrete-continuous function $f$ with respect to its input. Using the chain rule (and with a slight abuse of notation to

---

[1]For the sake of simplicity we assume the same input and output dimensions.

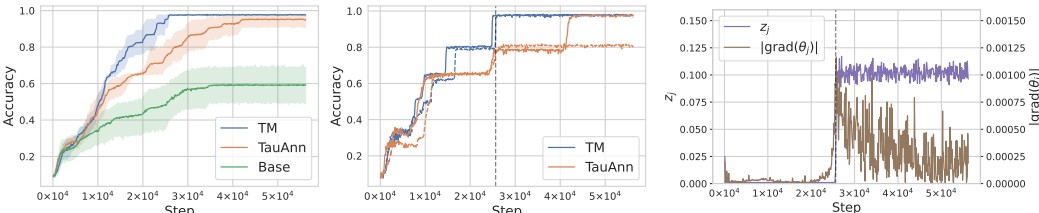

Figure 4: A comparison of the learning behavior of different annealing schemes for the MNIST addition task, averaged over 8 trained models. Left: The base model (Base), with constant parameters $\tau = 8$ and $\beta = 1$, cannot achieve an accuracy of more than 0.7. Increasing $\beta$ during training (TM) achieves better results than annealing $\tau$ (TauAnn). Middle: Two training runs for TM and TauAnn, respectively, where an accuracy of 0.9 was reached the latest. The accuracy curves of TM jump to the next plateau earlier and more consistently. The dotted grey line depicts the moment when a run of TM achieves an accuracy of $\sim 1.0$. Right: For the same run, $z_j$ and the absolute value of $\partial L/\partial \theta_j$ are plotted. The downstream gradient signal for $\theta_j$ is largely neutralized by a small value of $z_j$ until the corresponding category is used and its average probability abruptly reaches 0.1.

simplify the presentation), we can write $\partial \boldsymbol{v}/\partial \boldsymbol{u} = (\partial \boldsymbol{v}/\partial \boldsymbol{z})(\partial \boldsymbol{z}/\partial \boldsymbol{\theta})(\partial \boldsymbol{\theta}/\partial \boldsymbol{u})$. By assumption $\partial \boldsymbol{v}/\partial \boldsymbol{z}$ and $\partial \boldsymbol{\theta}/\partial \boldsymbol{u}$ exist and are efficiently computable. Using the derivative of the softmax function, we have

$$\frac{\partial z_i}{\partial \theta_j} = \begin{cases} \frac{1}{\tau} z_i(1 - z_i) & \text{if } i = j \\ -\frac{1}{\tau} z_i z_j & \text{if } i \neq j. \end{cases} \tag{5}$$

We have empirically observed that during training, there is a tendency of the models to not utilize all existing categories of the distribution and to wrongfully map different input representations to the same category. For instance, for the MNIST addition problem, all encoded images of two digits are mapped to the same category of the distribution. In these cases, the gradients of the parameters of the unused categories are vanishingly small. In order to analyze this behavior more closely, we derive an upper bound on the gradient of a parameter $\theta_j$ of the categorical distribution with respect to a loss function $L$. First, we have that

$$\left\| \left( \frac{\partial \boldsymbol{z}}{\partial \theta_j} \right) \right\|_F^2 = \sum_i \left| \frac{\partial z_i}{\partial \theta_j} \right|^2 = \frac{1}{\tau^2} z_j^2 \big( (1 - z_j)^2 + \sum_{i \neq j} z_i^2 \big) \leq \frac{2}{\tau^2} z_j^2, \tag{6}$$

where $\|\cdot\|_F$ is the Frobenius norm. The full derivation of this inequality can be found in the Appendix. Since the Frobenius norm is a sub-multiplicative matrix norm, we can write

$$|\text{grad}(\theta_j)| = \left\| \frac{\partial L}{\partial \theta_j} \right\|_F = \left\| \frac{\partial L}{\partial \boldsymbol{z}} \frac{\partial \boldsymbol{z}}{\partial \theta_j} \right\|_F \leq \left\| \left( \frac{\partial L}{\partial \boldsymbol{z}} \right) \right\|_F \left\| \left( \frac{\partial \boldsymbol{z}}{\partial \theta_j} \right) \right\|_F \leq \frac{\sqrt{2}}{\tau} \left\| \left( \frac{\partial L}{\partial \boldsymbol{z}} \right) \right\|_F z_j \tag{7}$$

where the last inequality follows from eq. (6). Hence, a small value of $\theta_j$ (and consequently $z_j$) leads to a small gradient for $\theta_j$. As a consequence, such models are more prone to fall into poor minima during training. Figure 4 illustrates an example of such a situation, which we encountered in practice. The small value of a specific $z_j$ leads to a small gradient at $\theta_j$ (right) as well as to suboptimal plateauing of the accuracy curve (middle). In other words, the downstream gradients information for $\theta_j$ is neutralized by a small $z_j$. Note that this issue only occurs when $L$ is more complex and not the categorical-cross entropy loss.

Another related problem is saturated probabilities in sequentially connected probabilistic components. Similar to the problem of sigmoid activation units in deep neural networks, which have been replaced by ReLUs and similar modern activation functions, saturated probabilities lead to vanishing gradients. Indeed, we observe in several experiments that the Gumbel-softmax distributions saturate and, as a result, that sufficient gradient information cannot reach the parameters of upstream neural network components.

We propose two strategies to mitigate the vanishing gradient behavior in complex discrete-continuous computation graphs. First, we analyze the interplay between the temperature parameter $\tau$ and the scale parameter $\beta$ of the Gumbel distribution. We make the subtle difference between these parameters explicit, which may also be of interest for improving stochastic softmax tricks [26]. By increasing

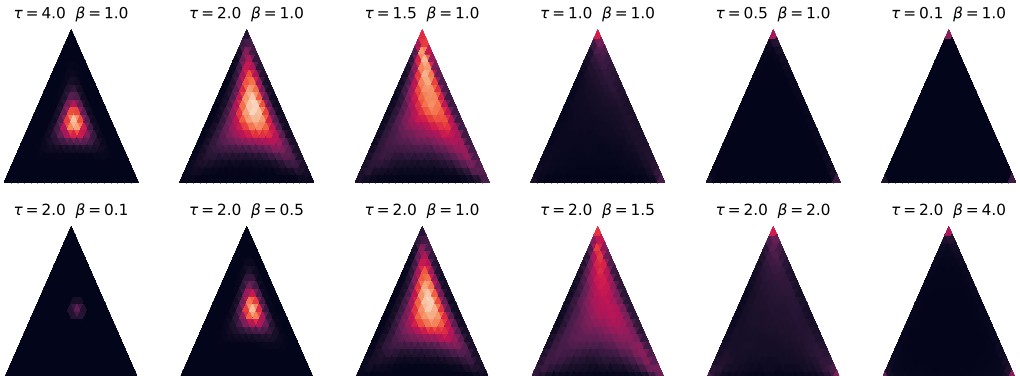

Figure 5: Two annealing schemes for the Gumbel-softmax trick for $\theta = (2, 0.5, 1)$. Top: The standard procedure of annealing the softmax temperature $\tau$ as introduced in [19] (see eq. (4)). Bottom: Starting with $\beta = 0$ and increasing it during training, the samples are increasingly more discrete but, compared to the standard annealing approach ($\tau \to 0$), more uniformly distributed over the corners of the probability simplex.

$\beta$ relative to $\tau$ while keeping $\tau$ fixed, we increase the probability of a gradient flow in the case of saturation. We show that the larger $\beta$, the more uniformly distributed (over the classes) are the discrete samples from the distribution, increasing the chance to escape poor minima. Second, we introduce DROPRES connections allowing us to lower-bound the gradients in expectation and leading to more direct gradient flow to parameters of upstream model components.

## 2.1 TEMPMATCH: Temperature Matching

We explore the behavior of two interdependent parameters: the Gumbel-softmax temperature $\tau$ and the Gumbel scale parameter $\beta$. First, we have the temperature parameter $\tau$ from the Gumbel-softmax trick (see eq. (4)). The purpose of this parameter is to make the output of the softmax function more or less discrete, that is, more or less concentrated on one of the categories. Second, and this is a new insight, we can adjust the scale parameter $\beta$ of the Gumbel distribution. For scale parameter $\beta$, we sample the noise perturbation iid as $\epsilon_i \sim \text{Gumbel}(0, \beta)$. If we use the Gumbel-max trick of eq. (3) we implicitly generate samples from the distribution $p(\boldsymbol{z}_i; \boldsymbol{\theta}) = \exp(\boldsymbol{\theta}_i/\beta) / \sum_{j=1}^{k} \exp(\boldsymbol{\theta}_j/\beta)$. Increasing the scale parameter $\beta$, therefore, makes the Gumbel-max samples more uniform and less concentrated. When using the Gumbel-softmax trick instead of the Gumbel-max trick, we obtain samples $\boldsymbol{z}_i$ that are more uniformly distributed over the categories and more discrete. Indeed, in the limit of $\beta \to \infty$, we obtain discrete samples uniformly distributed over the categories, independent of the logits. For $\beta \to 0$, we obtain the standard softmax function. Figure 5 illustrates the impact of the two parameters on the Gumbel-softmax distribution. Now, the problem of insufficient gradients caused by local minima can be mitigated by increasing the scale parameter $\beta$ *relative to* the temperature parameter $\tau$. The annealing schedule we follow is the inverse exponential annealing $\beta_t = \tau(1 - e^{-t\gamma})$ for some $\gamma > 0$. Increasing $\beta$ increases the probability of generating samples whose maximum probabilities are more uniformly distributed over the categories. This has two desired effects. First, samples drawn during training have a higher probability of counteracting poor minima caused at an earlier stage of the training. Figure 4 (left) shows that higher values for $\beta$ allow the model to find its way out of poor minima. A higher value of $\beta$ makes the model more likely to utilize categories with small $\theta$s, allowing the model to obtain improved minima (Figure 4 (middle)). Second, gradients propagate to parameters of the upstream components even after downstream components are saturating. To illustrate the second effect, we conducted the toy experiment depicted in Figure 2. The model here computes $\text{Softmax}(\boldsymbol{\theta} + \boldsymbol{\epsilon})$ with parameters $\boldsymbol{\theta}^\intercal = (\theta_1, \theta_2)$ and $\boldsymbol{\epsilon} \sim \text{Gumbel}(0, \beta)$. The learning problem is defined through a cross-entropy loss between the output probabilities and a constant target vector $(1.0, 0.0)^\intercal$. We observe that early during training, lower values for $\beta$ work well and exhibit less variance than higher values. However, once the probabilities and, therefore, the gradients start to saturate, a higher value for $\beta$ enables the continued training of the model. While the example is artificial, it is supposed to show that larger values of the scale parameter $\beta$, sustain gradients for upstream components even if downstream components have saturated.

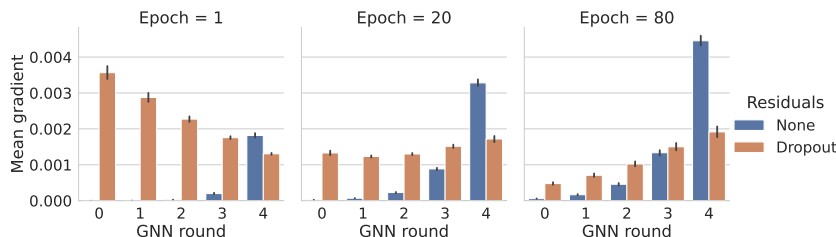

Figure 6: The influence of dropout residuals on the mean absolute gradients for ListOps. The values on the x-axis are the GNN message-passing iterations: the higher the number, the closer to the loss function is the corresponding discrete-continuous component. Adding dropout residual connections mitigates the vanishing gradient problem for components with greater distance to the loss.

In summary, increasing the Gumbel scale parameter $\beta$ has positive effects on the training dynamics. We propose an exponential increasing scheme relative to $\tau$. We show empirically that for more realistic learning problems choosing a higher value for $\beta$ is beneficial, especially later during training.

## 2.2 DROPRES: Residual Dropout Connections

Residual (or shortcut) connections are crucial when training deep neural networks [11]. A standard residual connection for the discrete-continuous components we consider here would be achieved by replacing Equation (2d) with $v = u + h_{w'}(z)$. By creating a direct connection to the continuous input of the discrete distributions, the optimizing problem simplifies. Unfortunately, what sets our problem apart from other end-to-end learnable neural networks is that we want to completely remove the residual connections at test time to obtain pure discrete representations and, therefore, the desired generalization properties. To mitigate the problem of overfitting to the existence of residual connections while at the same time reducing vanishing gradients in expectation, we propose DROPRES connections. These are residual connections sampled from a Bernoulli distribution with parameter $\alpha$. We replace Equation (2d) by

$$v = \begin{cases} u + h_{w'}(z) & \text{with probability } 1 - \alpha \\ h_{w'}(z) & \text{with probability } \alpha, \end{cases} \tag{8}$$

that is, we drop out the residual connection with probability $\alpha$. With probability $(1 - \alpha)$ we obtain a *shortcut connection* between the neural components, effectively bypassing the Gumbel-softmax. With probability $\alpha$ the training path goes exclusively through the Gumbel-softmax distribution. The expectation of the gradients, taken over a Bernoulli random variable with parameter $\alpha$, is now

$$\mathbb{E}\left[\frac{\partial v}{\partial u}\right] = (1 - \alpha)\mathbf{1} + \frac{\partial v}{\partial z}\frac{\partial z}{\partial \theta}\frac{\partial \theta}{\partial u}. \tag{9}$$

We propose a simple linear scheme to modify $\alpha$ from $0 \to 1$ during training. Hence, the model obtains a stronger gradient signal for upstream models *in expectation* in the early phase of training.

To illustrate the impact of dropout residual connections, we analyzed the gradients of message passing steps for the ListOps problem (see experiments) in a discrete-continuous computation graph of the type depicted in Figure 1 (bottom). As we can observe in Figure 6, if we do not use dropout residual connections, the greater the distance of a discrete-continuous operation to the loss function (the lower the value on the x-axis), the smaller is the mean absolute value of the gradients reaching it. This illustrates the vanishing gradient problem. When using dropout residual connections, on the other hand, the mean absolute values of the gradients do not vanish proportional to their distance to the loss and are more evenly distributed.

To summarize, optimizing the parameters of models comprised of discrete distributions and continuous neural network components is challenging, mainly due to local minima and vanishing gradients. By setting the Gumbel scale parameter $\beta = 0$ as well as the dropout probability of the residual connections $\alpha = 0$ at the beginning of training, we obtain a continuous and deterministic relaxation of the target model. Increasing the DROPRES parameter $\alpha$ over time makes the model increasingly use the discrete stochastic nodes. Increasing the Gumbel scale parameter $\beta$ allows the model to escape local minima and small gradients.

## 3 Related Work

Various gradient estimators have been proposed for the problem of backpropagating through stochastic nodes. The REINFORCE algorithm [40] utilizes the log derivative trick. The Straight-Through estimator [2] back-propagates through hard samples by replacing the threshold function by the identity in the backward pass. More recent approaches are based on reparameterization tricks [15] that enable the gradient computation by removing the dependence of the density on the input parameters. Maddison et al. [19] and Jang et al. [14] propose the Gumbel-Softmax trick, a continuous (but biased) reparameterization trick for categorical distributions. Tucker et al. [38] and Grathwohl et al. [9] introduce parameterized control variates to lower the variance for these gradient estimators. They show that these estimators can, in theory, also be generalized to chains of stochastic nodes but do not discuss any learning behavior. Shayer et al. [32] modify the local reparameterization trick [16] to improve the training of discrete gradient estimators. Recently, Paulus et al. [26] proposed a framework that generalizes the Gumbel-Softmax trick for various discrete probability distributions. Our aim is not to improve gradient estimators for a single stochastic node but to analyze the behavior of discrete-continuous computation graphs with multiple sequential discrete components. Pervez et al. [27] utilize harmonic analysis for Boolean functions to control the bias and variance for gradient estimates for Boolean latent variable models. We focus on efficiently training models with gradient estimates for categorical distributions. Huang et al. [13] propose to drop out full layers of ResNet to obtain models of stochastic depth during training. In contrast, we use dropout on the residual connections to approximate the associated discrete model. SparseMAP [24] is an approach to structured prediction and latent variables, replacing the exponential distribution (specifically, the softmax) with a sparser distribution. LP-SparseMAP [23] is a continuation of SparseMAP using a relaxation of the optimization problem rather than a MAP oracle. In addition, the idea to enforce more sparsity can also be exploited for efficient marginal inference in latent variable models [5] which can then be used to compute stochastic gradients.

The respective advantages of neural and symbolic methods have led to various attempts to combine these two. Early work like Towell and Shavlik [35] and Hölldobler et al. [12] have explored the possibilities of translating propositional logic programs into neural networks. Evans and Grefenstette [8] and Dong et al. [7] build upon these ideas by utilizing the framework of inductive logic programming to learn rules from examples. Kool et al. [18] combine an attention model and the REINFORCE algorithm with a greedy baseline to learn heuristics for combinatorial optimization problems. A different direction of work focuses on knowledge base reasoning. Rocktäschel and Riedel [28] combine Prolog's backward chaining algorithm with a differentiable unification whereas Yang et al. [41] learn to compose the inference task into differentiable operations. Serafini and Garcez [31] propose a differentiable first-order logic language to differentiate deductive reasoning, and Manhaeve et al. [20] extend the probabilistic logical language ProbLog to interact with neural networks. Also related is prior work on automatic program induction. Andreas et al. [1] combine neural networks with the compositional linguistic structure of questions, Mao et al. [21] additionally learn visual concepts and Tsamoura and Michael [37] introduce the concept of abduction to learn the inputs of a symbolic program. All prior work integrates neural modules into classical or fuzzy logic or separates the neural and the symbolic parts in other ways. In contrast, we propose a method that allows training complex discrete-continuous computation graphs with multiple sequential discrete distributions.

## 4 Experiments

The aim of the experiments is threefold. First, we want to analyze the behavior of complex stochastic computations graphs arising in typical application domains such as multi-hop reasoning in knowledge graphs and unsupervised parsing of lists of operations. Second, we want to evaluate the behavior of the stochastic computation graphs when incorporating the proposed methods (dropout residual connections and temperature matching) to improve the vanishing gradient problem. Third, we want to compare the resulting discrete-continuous models with state of the art models which do *not* have stochastic components. Here, we are especially interested in analyzing the generalization (extrapolation) behavior of the models under consideration. The implementations are in PyTorch and can be found at `https://github.com/nec-research/dccg`. All experiments were run on a GeForce RTX 2080 Ti GPU.

Table 1: The results for unsupervised parsing on ListOps. Results taken from Paulus et al. [26] are marked with an asterisk (*) and are based on a different (unpublished) dataset.

| | | | | Task acc. (extrapolation) | |
| Model | Task acc. | Edge prec. | Inter. acc. | $d = 8$ | $d = 10$ |
| --- | --- | --- | --- | --- | --- |
| Und.* | $91.2 \pm 1.8^*$ | $33.1 \pm 2.9^*$ | - | n.a. | n.a. |
| Arb.* | $95.0 \pm 3.0^*$ | $75.0 \pm 7.0^*$ | - | n.a. | n.a. |
| LSTM | $91.5 \pm 0.3$ | - | - | $83.7 \pm 2.0$ | $76.9 \pm 3.7$ |
| Arb., $\tau = 2$ | $\mathbf{96.8} \pm 0.3$ | $77.4 \pm 1.8$ | - | $84.3 \pm 1.4$ | $75.4 \pm 1.7$ |
| Ours, $\tau = 1$ | $96.1 \pm 0.4$ | $\mathbf{82.3} \pm 1.1$ | $\mathbf{70.9} \pm 1.2$ | $92.6 \pm 1.1$ | $86.9 \pm 4.9$ |
| Ours, $\tau = 2$ | $\mathbf{96.3} \pm 0.5$ | $76.8 \pm 2.2$ | $62.9 \pm 2.4$ | $\mathbf{92.7} \pm 1.3$ | $\mathbf{88.7} \pm 3.3$ |
| Arb. (GT) | $98.7 \pm 0.1$ | $100.0$ | - | $86.4 \pm 1.1$ | $71.0 \pm 2.1$ |
| Ours (GT) | $99.8 \pm 0.1$ | $100.0$ | $99.9 \pm 0.0$ | $99.8 \pm 0.1$ | $99.9 \pm 0.1$ |

Table 2: The ablation study for the ListOps task. Temperature matching improves the training behavior of the model. The addition of dropout residual connections is crucial for efficient learning.

| | | | | Task acc. (extrapolation) | |
| Model | Task acc. | Edge prec. | Inter. acc. | $d = 8$ | $d = 10$ |
| --- | --- | --- | --- | --- | --- |
| Ours, $\tau = 1$ | $96.1 \pm 0.4$ | $82.3 \pm 1.1$ | $70.9 \pm 1.2$ | $92.6 \pm 1.1$ | $86.9 \pm 4.9$ |
| (-) TM | $95.1 \pm 1.7$ | $76.9 \pm 12.1$ | $66.3 \pm 9.2$ | $90.5 \pm 2.4$ | $83.7 \pm 4.0$ |
| (-) DR | $74.8 \pm 11.5$ | $43.9 \pm 30.3$ | $29.4 \pm 18.6$ | $70.5 \pm 9.3$ | $66.7 \pm 8.8$ |
| (-) DR, TM | $56.2 \pm 5.0$ | $15.1 \pm 7.7$ | $12.7 \pm 3.2$ | $53.1 \pm 6.6$ | $49.2 \pm 7.1$ |

**Unsupervised Parsing on ListOps** The Listops dataset contains sequences in prefix arithmetic syntax such as $\max[\,2\,9\,\min[\,4\,7\,]\,0\,]$ and its unique numerical solutions (here: 9) [22]. Prior work adapted and used this dataset to evaluate the performance of stochastic softmax tricks [26]. Following this prior work, we first encode the sequence into a directed acyclic graph and then run a graph neural network (GNN) on that graph to compute the solution. Since the resulting dataset was not published, we generated a dataset following their description. In addition to examples of depth $1 \leq d \leq 5$, we further generated test examples of depth $d = 8, 10$ for the extrapolation experiments. More details on the dataset creation can be found in the supplementary material.

The arithmetic syntax of each training example induces a directed rooted in-tree, from now on called arborescence, which is the tree along which the message-passing steps of a GNN operate. We use the same bi-LSTM encoder as Paulus et al. [26] to compute the logits of all possible edges and the same directed GNN architecture. The original model uses a directed version of Kirchoff's matrix-tree theorem as introduced by Koo et al. [17] to induce the arborescence prior. We simplify this idea by taking, for each node, the Gumbel-softmax over all possible parent nodes. Here we make use of the fact that in an arborescece, each non-root node has exactly one parent. Note that this prior is slightly less strict than the arborescence prior. As in Paulus et al. [26], we exclude edges from the final closing bracket $]$ since its edge assignment cannot be learned from the task objective.

In contrast to Paulus et al. [26], where only the edges of the latent graph are modeled with categorical variables, we also modelled the nodes of the latent graphs with discrete-continuous components (see Figure 1 (bottom)). Let $\mathrm{Num} \in \mathbb{R}^{10 \times \dim}$ be the embedding layer that maps the 10 numeral tokens to their respective embeddings and let $\mathrm{Pred} : \mathbb{R}^{\dim} \to \mathbb{R}^{10}$ be the classification layer that maps the embedding $\boldsymbol{x} \in \mathbb{R}^{\dim}$ of the final output to the logits of the 10 classes: $\mathrm{Pred}(\boldsymbol{x}) := \mathrm{Lin}^{10 \times \dim}(\mathrm{ReLU}(\mathrm{Lin}_B^{\dim \times \dim}(\boldsymbol{x})))$. After each message-passing operation of the GNN, we obtain an embedding $\boldsymbol{u} \in \mathbb{R}^{\dim}$ for each node. By choosing $g_{\boldsymbol{w}} := \mathrm{Pred}$, we obtain the 10 class logits $\boldsymbol{\theta} = g_{\boldsymbol{w}}(\boldsymbol{u}) = \mathrm{Pred}(\boldsymbol{u}) \in \mathbb{R}^{10}$ for the numerals $0, \ldots, 9$. Using the Gumbel-softmax trick with logits $\boldsymbol{\theta}$, scale $\beta$, and temperature $\tau$, we obtain $\boldsymbol{z} \in \mathbb{R}^{10}$. We then compute $\boldsymbol{v} = h_{\boldsymbol{w'}}(\boldsymbol{z}) = \boldsymbol{z}^{\mathsf{T}} \mathrm{Num}$. We apply this on all intermediate node embeddings simultaneously and repeat this after each of the first 4 out of 5 message-passing rounds of the GNN. We use dropout residual connection with increasing dropout probability $\alpha$. At test time, the model performs discrete and interpretable operations. Figure 1 depicts an example stochastic computation graph for the task. We used the same LSTM as in Paulus et al. [26] and a re-implemented version of their model with our customized prior. For ablation purposes, we run all our models with and without dropout residuals

Table 3: Results for the path query benchmark [10].

| Model | WordNet | | Freebase | |
|---|---|---|---|---|
| | MQ | H@10 | MQ | H@10 |
| Bilinear-C | 89.4 | 54.3 | 83.5 | 42.1 |
| DistMult-C | 90.4 | 31.1 | 84.8 | 38.6 |
| TransE-C | 93.3 | 43.5 | 88.0 | 50.5 |
| Path-RNN | **98.9** | – | – | – |
| ROP | – | – | 90.7 | 56.7 |
| CoKE | 94.2 | **67.9** | **95.0** | **77.7** |
| Ours, $\tau = 4$ | 94.4 | 64.3 | 89.6 | 68.7 |

Table 4: Results for the extrapolation benchmark [39] for the path query dataset [10].

| Paths | Model | WordNet | | Freebase | |
|---|---|---|---|---|---|
| | | MQ | H@10 | MQ | H@10 |
| $\leq 4$ | CoKE | 93.6 | **65.5** | 93.1 | **71.2** |
| | Ours | **94.3** | 64.7 | 88.8 | 69.4 |
| $\leq 3$ | CoKE | 92.6 | **65.0** | 90.6 | 64.6 |
| | Ours | **93.4** | **65.0** | 88.3 | **68.5** |
| $\leq 2$ | CoKE | **90.8** | 49.5 | **89.4** | 59.5 |
| | Ours | 90.2 | **62.0** | 87.6 | **68.6** |
| $\leq 1$ | CoKE | 73.5 | 16.7 | 72.7 | 37.3 |
| | Ours | **81.1** | **52.2** | **82.1** | **63.9** |

and temperature matching, respectively. All models are run for 100 epochs with a learning rate of 0.005 and we select $\tau \in \{1, 2, 4\}$. We keep all other hyperparameters as in Paulus et al. [26]. We evaluate the task accuracy, the edge precision (using the correct graphs), the task accuracy on intermediate operations, and two extrapolation tasks. For the evaluation of the extrapolation task with depth $d$, we run the GNN $d$ rounds instead of 5.

Table 1 compares the proposed models with state of the art approaches. The best performing models are those with discrete-continuous components trained with dropout residuals and temperature matching. While our models do not improve the task accuracy itself, they exceed state of the art methods on all other metrics. Most noticeable is the improved generalization behavior: our models extrapolate much better to expressions with a depth not seen during training. This becomes even more visible when trained on the ground truth graph directly (instead of on the sequence) where the generalization accuracy is close to 100%. Our model also achieves the best performance on the graph structure recovery with a precision of 82.3%. The mismatch between the task accuracy and the edge precision is mainly due to examples for which the correct latent graph is not required to solve the problem, e.g., for the task max[ 2 max[ 4 7 ]] it does not matter whether the 7 points to the first or the second max[. Note that our method performs discrete operations at test time which are therefore verifiable and more interpretable. Our model generates intermediate representations that can be evaluated in exactly the same way as the final outputs, see Column *Inter. acc.* in Table 1.

The ablation study in Table 2 highlights the challenge of learning discrete-continuous computation graphs with multiple sequential Gumbel-softmax components. It is entirely impossible to train the model without dropout residuals and temperature matching. The analysis in Section 2 Figure 6 reveal the reason. Sequential discrete distributions in the computation graph cause vanishing gradients. Temperature matching stabilizes learning somewhat but to avoid vanishing gradients entirely, the use of dropout residuals is necessary.

**Multi-Hop Reasoning over Knowledge Graphs** Here we consider the problem of answering multi-hop (path) queries in knowledge graphs (KGs) [10]. The objective is to find the correct object given the subject and a sequence of relations (the path) without knowing the intermediate entities. We evaluate various approaches on the standard benchmarks for path queries [10].[2]

We model each application of a relation type to an (intermediate) entity as a discrete-continuous component that discretely maps to one of the KG entities. The discrete distribution $p(\boldsymbol{z}; \boldsymbol{\theta})$, therefore, has as many categories as there are entities in the KG. Let $\mathrm{score}(\boldsymbol{s}, \boldsymbol{r}, \boldsymbol{o}) : \mathbb{R}^{3 \times \dim} \to \mathbb{R}$ be a scoring function that maps the embeddings of a triple to the logit (unnormalized probability) of the triple being true. Given the current (intermediate) entity embedding $\boldsymbol{s}$ and relation type embedding $\boldsymbol{r}$ we compute the logits $\boldsymbol{\theta}$ for all possible entities. Hence, the function $g$ computes the logits for all possible entities. Using the Gumbel-softmax trick with parameter $\boldsymbol{\theta}$, scale $\beta$, and temperature $\tau$, we obtain the sample $\boldsymbol{z} \in \mathbb{R}^n$. The function $h$ now computes $\boldsymbol{z}^\mathsf{T}\mathbf{E}$ where $\mathbf{E}$ be the matrix whose rows are the entity embeddings. We use dropout residual connection between the input and output vectors of the discrete-continuous component during training. Note that we do not increase the number of

---

[2]Revised results in Table 3 and Table 4 because of a flawed evluation protocal in the original results.

Table 5: A comparison of our proposed method with DeepProbLog and a CNN baseline.

| Model | Task Acc. | Interpretable | Discrete (test time) | Learned Structure |
|---|---|---|---|---|
| Baseline | $89.14 \pm 1.22$ | No | No | No |
| DeepProbLog | $98.17 \pm 0.20$ | Yes | Yes | No |
| Ours, $\tau = 8$ | $98.10 \pm 0.15$ | Yes | Yes | Yes |

parameters compared to the base scoring models. We use ComplEx [36] as the scoring function. We choose a dimension of 256, a batch size of 512 and learning rate of 0.001. We further train 1vsAll with the cross-entropy loss for 200 epochs and with a temperature of $\tau = 4$. We compare our best performing model on the original dataset from Guu et al. [10] with their proposed evaluation protocol and baselines against RNN models, Path-RNN [6], ROP [42] and the state-of-the-art transformer model CoKE [39] for the standard task as well as in an extrapolation setting.

Table 3 lists the results for the proposed model in comparison to several baselines. Our proposed model performs significantly better than the baselines KGC models which do not have stochastic components (labeled with -C). On WordNet, our model can even keep pace with the state of the art transformer model CoKE.

Similar to the results on ListOps, the discrete-continuous models have the strongest generalization performance (see Table 4). Here, we use the same test set as before but reduce the length of the paths seen during training. Even when trained only on paths of length $\leq 3$, our model performs better on the path query task than most baselines trained on paths of all lengths.

**End-to-End Learning of MNIST Addition**  The MNIST addition task addresses the learning problem of simultaneously (i) recognizing digits from images and (ii) performing the addition operation on the digit's numerical values [20]. The dataset was introduced with DeepProbLog, a system that combines neural elements with a probabilistic logic programming language. In DeepProbLog the program that adds two numbers is provided. We adopt their approach of using a convolutional neural network (CNN) to encode the MNIST images. Contrary to their approach, we learn the operation without any prior knowledge about adding numbers in a data-driven manner.

The addition operation is modeled as $\mathrm{Add}(\boldsymbol{x}, \boldsymbol{y}) = \mathrm{Lin}_B^{\dim \times 2\dim}(\mathrm{ReLU}(\mathrm{Lin}_B^{2\dim \times 2\dim}([\boldsymbol{x}; \boldsymbol{y}])))$ and we use a single linear layer to compute the logits. Given $\boldsymbol{u} \in \mathbb{R}^{\dim}$, the encoding of the digit obained from the CNN, the function $g$ computes $\boldsymbol{\theta} = \mathrm{Lin}(\boldsymbol{u})$, that is, the 19 logits for all possible numbers. Using the Gumbel-softmax trick with parameter $\boldsymbol{\theta}$, scale $\beta$, and temperature $\tau$, we obtain $\boldsymbol{z} \in \mathbb{R}^{19}$. The function $h$ now computes $\boldsymbol{v} = \boldsymbol{z}^{\intercal} \mathrm{P} \in \mathbb{R}^{\dim}$ where P is the weight matrix of Lin. We discretize the two outputs of the CNN, execute the addition layer and use the single linear layer P for the classification. This results into a model that has in total fewer parameters than the baseline in [20]. We use a learning rate of 0.0001 and a temperature of $\tau = 8$.

Table 5 lists the results on this task. Our proposed model, even though trained without a given logic program and in a data-driven manner, achieves the same accuracy as DeepProbLog. Since we use hard sampling during test time, we can discretely read out the outputs of the internal CNN layer.

## 5   Conclusion

We introduced two new strategies to overcome the challenges of learning complex discrete-continuous computation graphs. Such graphs can be effective links between neural and symbolic methods but suffer from vanishing gradients and poor local minima. We propose dropout residual connections and a new temperature matching schema for mitigating these problems. Note that even with our proposed methods, learning models with discrete nodes is still considerably more challenging than standard neural networks. Furthermore, it is not clear how to use dropout residuals for structural stochastic nodes or in such setups in which only hard decisions are feasible. In experiments across a diverse set of domains, we demonstrate that our methods enable the training of complex discrete-continuous models, which could not be learned before. These models beat prior state of the art models without discrete components on several benchmarks and show a remarkable gain in generalization behavior. Promising future work includes applying the proposed methods to reinforcement learning problems, effectively sidestepping the problem of using the high variance score function estimator.

# 6 Acknowledgements

Many thanks to the NeurIPS reviewers for their highly constructive feedback and suggestions. Their input has helped to improve the paper tremendously. Special thanks to reviewer RKc5 who provided an impressive review and engaged in a long and fruitful discussion with us.

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
