# 7 Appendix

## 7.1 Implementation Details

All our methods were implemented with PyTorch and were run on a GeForce RTX 2080 Ti GPU. For the experiments we picked the best performing learning rate out of $\{1e^{-4}, 3e^{-4}, 5e^{-4}, 1e^{-3}\}$ and the best performing softmax temperature $\tau$ out of $\{1, 2, 4\}$. We found a temperature of $\tau = 4$ not sufficiently high enough for the MNIST addition experiments and picked $\tau = 8$ for these experiments.

### 7.1.1 Unsupervised Parsing on ListOps

For all ListOps experiments, we use a hidden dimension $\dim = 60$, a batch size of $100$, and train the model for a total number of $100$ epochs. We use the Adam optimizer with a learning rate of $0.0005$ to minimize the cross-entropy loss. After every epoch, we evaluate on the validation set and save the best model. We use discretization after the first $4$ out of the $5$ GNN message passes. We always use an exponential function to increase $\beta$ and a linear function to increase $\alpha$ (during experiments for which they are not constant). The temperatures are updated $10$ times per epoch; we use $\beta_t = \tau(1 - e^{-t\gamma})$ for $\gamma = 0.008$, $\tau = 1$ and $\alpha_t = \max(1, 0.002t)$. We repeat every experiment $8$ times.

Our model is taken from Paulus et al. [26] and has following structure. For the encoder, we start with an embedding layer for the $14$ tokens. We use two independent one-directional LSTMs with a single layer, respectively. We use a token-wise multiplication of the two sequence outputs to obtain a latent graph and use Gumbel-softmax on this latent graph:

$$
\begin{aligned}
x &= E_1[14, \dim](tok) \\
q &= LSTM_1(x) \\
k &= LSTM_2(x) \\
A'_{ij} &= qk^\intercal \\
A_{ij} &= \frac{1}{\lambda}(A'_{ij} + \mathrm{Gumbel}(0, \tau)).
\end{aligned}
$$

To obtain the edge precision, we compare the latent graph $A_{ij}$ with the ground truth adjacency matrix $B_{ij}$. Due to our arborescence prior as described in the main paper, we always have the same number of edges resulting in equality of edge precision and edge recall. For the GNN we use a second, independent embedding layer for the tokens. For each GNN message pass, we use the latent representation after the embedding layer of a token and concatenate it with the representation of the node of the incoming message. In the baseline experiments (Arb.), we use the current state of the token instead of the first embedding during all message passes except the first one. The message is transformed by a message MLP with dropout probability $0.1$ and is summed up regarding the edge weights of the latent graph $A^{ij}$. The new node state is summed with the one before the message pass:

$$
\begin{aligned}
x_e &= E_2[14, \dim](tok) \\
p'' &= [x_e; x_j] \quad \text{or} \quad [x_i; x_j] \\
p' &= \mathrm{Dropout}(\mathrm{ReLU}(\mathrm{Lin}_{1,Bias}^{\dim \times 2\dim}(p''))) \\
p &= \mathrm{ReLU}(\mathrm{Lin}_{2,Bias}^{\dim \times \dim}(p')) \\
m &= A_i p \\
x'_i &= x_i + m.
\end{aligned}
$$

In those experiments that utilize the ground truth edges (GT) instead of the latent graph, we replace the second last line $m = A_i p$ by $m = B_i p$ with $B$ being the ground truth adjacency matrix. To create Figure 6, we read out the mean absolute values of the gradient at $x_e$ as well as $x'_i$ for each of the first $4$ GNN massage passes. For the discretization, we use the classification layer that maps the embedding of the final output to the logits of the $10$ classes and the weights of the embedding layer

that map the 10 numeral tokens to their respective embeddings:

$$\boldsymbol{\theta} = \mathrm{Lin}_4^{10 \times \dim}(\mathrm{ReLU}(\mathrm{Lin}_{3,Bias}^{\dim \times \dim}(x_i')))$$

$$\boldsymbol{z} = \frac{1}{\lambda}(\boldsymbol{\theta} + \mathrm{Gumbel}(0, \tau))$$

$$\boldsymbol{v} = \boldsymbol{z}^{\mathsf{T}} E_2([0, \ldots, 9])$$

$$x_i^{\text{next}} = \begin{cases} \boldsymbol{v} & \text{with probability } \alpha, \\ x_i' + \boldsymbol{v} & \text{else.} \end{cases}$$

To read out the intermediate results, we compare the final (after the last GNN message pass) $\boldsymbol{\theta}$ of all operation tokens with the ground truth intermediate results obtained by executing the operations. The experiments without dropout residuals are always set to utilize $\alpha = 1$. To obtain the task class of the full example, we utilize the same MLP from the discretization as the classification layer, i.e., $\mathrm{Lin}_4^{10 \times \dim}(\mathrm{ReLU}(\mathrm{Lin}_{3,Bias}^{\dim \times \dim}(x_0^{\text{last}})))$.
The baseline LSTM model consists of an one-directional LSTM with a single layer. The hidden state of the final token is fed to a classification MLP, i.e., $\mathrm{Lin}^{10 \times \dim}(\mathrm{ReLU}(\mathrm{Lin}_{Bias}^{\dim \times \dim}(h_{-1})))$.

### 7.1.2 Multi-Hop Reasoning over Knowledge Graphs

For all Knowledge Graph experiments, we use a hidden dimension $\dim = 256$, a batch size of $512$, and train the model for a total of 200 epochs. We use the Adam optimizer with a learning rate of $0.001$ to minimize the cross-entropy loss. We use a randomized grid search training on paths of length 1 and validating hitset 10 on paths of length 2 for the L2 regularization of the entities and the relations between $1e^{-20}, ..., 1e^{-5}$ and for the dropout probabilities for the subject, object and relations between $0, ..., 0.8$, respectively. This results in an entity regularization of $1e^{-15}$, a relation regularization of $1e^{-9}$, a subject dropout of 0.7 [0.1], an object dropout of 0.1 [0.6] and a relation dropout of 0.5 [0.2] for WordNet [Freebase]. After every 10th epoch, we evaluate on the validation set and save the best model. We use discretization after every single relation. We always use an exponential function to increase $\beta$ and a linear function to increase $\alpha$ (during experiments for which they are not constant). The temperatures are updated 3 times per epoch; we use $\beta_t = \tau(1 - e^{-t\gamma})$ for $\gamma = 0.008$, $\tau = 1$ and $\alpha_t = \max(1, 0.005t)$. We repeat every experiment 4 times.

Our model is based on the ComplEx model from Trouillon et al. [36]. Given the current (intermediate) entity embedding $\boldsymbol{s}$ and relation type embedding $\boldsymbol{r}$ we compute the logits $\boldsymbol{\theta}$ for all possible entities with the ComplEx scoring function $\mathrm{score} = \mathrm{Re} < \boldsymbol{s}, \boldsymbol{r}, \cdot >$. Hence, we obtain the logits for all possible entities. Using the Gumbel-softmax trick with parameter $\boldsymbol{\theta}$, scale $\beta$, and temperature $\tau$, we obtain the sample $\boldsymbol{z} \in \mathbb{R}^n$. The function $h$ now computes $\boldsymbol{z}^{\mathsf{T}}\mathbf{E}$ where $\mathbf{E}$ be the matrix whose rows are the entity embeddings. We use dropout residual connections between the input and output vectors of the discrete-continuous component during training. Guu et al. [10] introduced their own evaluation protocol for multi-hop reasoning that we adopted. On the one hand, we calculate all possible objects that can be reached traversing each path. These are the positives and are filtered during evaluation. On the other hand, we calculate all possible objects that can be reached by the final relation of the path individually. These are the negatives that we rank our prediction against. We compare our best performing model against two RNN models, Path-RNN [6] and ROP [42] as well as against the state-of-the-art transformer model CoKE [39].

We use a slightly different setup for the FB15K237 experiments in Section 7.4.1. Here, we train the same model for a total number of 100 epochs and for each path from subject to object as well as from object to subject using reciprocal relations [29]. The evaluation is copied from the standard link prediction task [3], that is, we evaluate all paths in the forward direction from subject to object as well as in the backward direction from object to subject. We also use the filter method for the positives, but we compare against all other possible objects and not only the ones reachable by the last relation.

### 7.1.3 End-to-End Learning of MNIST Addition

For the MNIST addition experiments, we use a batch size of 16 and train the model for a total number of 30 epochs. We use the Adam optimizer with a learning rate of $0.0001$ to minimize the cross-entropy loss. We use the dataset from Manhaeve et al. [20]. Since they do not offer a validation set, we validate the model twice per epoch on the test set and record the best test accuracy for all models. We use discretization after the CNN encoding layer, i.e., before the addition layer. The temperatures

are updated 8 times per epoch; we increase $\beta$ and $\alpha$ by $\beta_t = \tau(1 - e^{-t\gamma})$ for $\gamma = 0.008, \tau = 8$ and $\alpha_t = \max(1, 0.002t)$. We repeat every experiment 8 times.

Our model is based on the baseline model used by Manhaeve et al. [20] and has the following structure. The CNN encoder consists of two convolutional layers with kernel size 5 and filter size 6 and 16, respectively. Each convolutional layer is followed by 2D max-pooling layer of size $2 \times 2$ and a ReLU activation function. An MLP with 3 layers transforms the output to an embedding size of 84.

$$e' = \text{ReLU}(\text{maxpool}_{2\times2}(\text{Conv2d}_{6,5}(inp)))$$
$$e = \text{ReLU}(\text{maxpool}_{2\times2}(\text{Conv2d}_{16,5}(e')))$$
$$x' = \text{Lin}_{Bias}^{84\times84}(\text{ReLU}(\text{Lin}_{Bias}^{84\times120}(\text{ReLU}(\text{Lin}_{Bias}^{120\times256}(e))))).$$

For the discretization we use a single classification matrix $C \in \mathbb{R}^{19\times84}$. This matrix is also used for the classification after the addition layer.

$$\boldsymbol{\theta} = Cx'$$
$$\boldsymbol{z} = \frac{1}{\lambda}(\boldsymbol{\theta} + \text{Gumbel}(0, \tau))$$
$$\boldsymbol{v} = \boldsymbol{z}^{\mathsf{T}}C$$
$$x = \begin{cases} \boldsymbol{v} & \text{with probability } \alpha, \\ x' + \boldsymbol{v} & \text{else.} \end{cases}$$

The addition layer concatenates the final embeddings of both images and transform them through a MLP with 2 layers into a single representation.

$$x_{1,2} = \text{ReLU}(\text{Lin}_{Bias}^{84\times168}(\text{ReLU}(\text{Lin}_{Bias}^{168\times168}([x_1; x_2])))).$$

We obtain the final class log-probabilities by computing $Cx_{1,2}$. This results in a model with a total of $94,900$ parameters. We use the code offered by Manhaeve et al. [20] to run the DeepProbLog experiments as well as the baseline model they compared to. These two models are executed in a single epoch and use a batch size of 1 and 2, respectively. The DeepProbLog model uses a similar encoder to get predictions for each of the images individually and has the following structure.

$$e' = \text{ReLU}(\text{maxpool}_{2\times2}(\text{Conv2d}_{6,5}(inp)))$$
$$e = \text{ReLU}(\text{maxpool}_{2\times2}(\text{Conv2d}_{16,5}(e')))$$
$$out = \text{Lin}^{10\times120}(\text{ReLU}(\text{Lin}_{Bias}^{120\times256}(e))),$$

The 2 image outputs are then fed into the probabilistic logic program ProbLog together with the following annotated disjunction, which handles the logic of addition to obtain the final predictions.

$$\text{nn(mnist\_net}, [X], Y, [0, 1, 2, 3, 4, 5, 6, 7, 8, 9]) :: \text{digit}(X, Y);$$
$$\text{add}(X, Y, Z) : - \text{digit}(X, X2), \text{digit}(Y, Y2), Z \text{ is } X2 + Y2.$$

Different to DeepProbLog or our model, the baseline model concatenates the images beforehand and uses the following layers.

$$e' = \text{ReLU}(\text{maxpool}_{2\times2}(\text{Conv2d}_{6,5}(inp)))$$
$$e = \text{ReLU}(\text{maxpool}_{2\times2}(\text{Conv2d}_{16,5}(e')))$$
$$out = \text{Lin}^{19\times84}(\text{ReLU}(\text{Lin}_{Bias}^{84\times120}(\text{ReLU}(\text{Lin}_{Bias}^{120\times704}(e))))),$$

which results in a model with a total of $98,951$ parameters.

For the experiments in Figure 4, we also use a batch size of 16, set $\tau = 8.0$, use a learning rate of 0.0003 and train the model for a total number of 30 epochs. If we update temperatures, we also update them 8 times per epoch. We do not use dropout residuals. For the experiment Base, we use a constant $\tau = 8.0$, $\beta = 1.0$ and $\gamma = 0.008$. For the experiment TauAnn, we set $\beta = 1.0$ constant and anneal $\tau$ by $\tau_t = \max(1.0, \tau e^{-t\gamma})$. For the experiment TM, we set $\tau = 8.0$ constant and increase $\beta$ by $\beta_t = \tau(1 - e^{-t\gamma})$.

### 7.1.4 Temperature Matching

To illustrate the effect of the parameter $\beta$, we conducted two toy experiments. The first one is depicted in Figure 5. We use the same setup as in Maddison et al. [19]. We illustrate the Gumbel-softmax densities of the unnormalized probabilities $\theta = (2, 0.5, 1)$. The second toy experiment is depicted in Figure 2. The model here computes $\text{Softmax}(\boldsymbol{\theta} + \boldsymbol{\epsilon})$ with parameters $\boldsymbol{\theta}^\mathsf{T} = (\theta_1, \theta_2)$ and $\boldsymbol{\epsilon} \sim \text{Gumbel}(0, \beta)$. The learning problem is defined through a cross-entropy loss between the output probabilities of the Gumbel-Softmax and a constant target vector $(1.0, 0.0)^\mathsf{T}$. We use the initial paramaters $\boldsymbol{\theta}^\mathsf{T} = (-0.9442, 0.3893)$ and minimize the loss using stochastic gradient descent with a learning rate of $0.01$ for a total number of $5000$ steps.

### 7.2 Datasets

For the **Unsupervised Parsing on ListOps** experiment, we followed the data generation description from Paulus et al. [26] applied to the original code from Nangia and Bowman [22]. More precisely, we used the three operators $\min$, $\max$ and $\text{med}$ and capped the maximum length of a sequence at $50$. For each depth $d \in \{1, 2, 3, 4, 5\}$ we generated $20,000$ samples for the training set and $2,000$ samples each for the validation set and test set. In order to evaluate the generalization behavior, we further generated $2,000$ test samples for each depth $d \in \{8, 10\}$. All other settings are those of the original code [22].

### 7.3 Derivation of Equation 6

For all $i$, we have $z_i \in [0, 1]$ and $\sum_i z_i = 1$. Therefore, we have $\sum_{i \neq j} z_i^2 \leq 1$ and $(1 - z_j)^2 \leq 1$ and thus,

$$
\begin{aligned}
\left\| \left( \frac{\partial \boldsymbol{z}}{\partial \theta_j} \right) \right\|_F &= \left\| \left( \frac{\partial z_1}{\partial \theta_j} \cdots \frac{\partial z_n}{\partial \theta_j} \right)^\mathsf{T} \right\|_F \\
&= \sqrt{\sum_i \left| \frac{\partial z_i}{\partial \theta_j} \right|^2} \\
&= \sqrt{\left| \frac{1}{\tau} z_j (1 - z_j) \right|^2 + \sum_{i \neq j} \left| \frac{1}{\tau} z_i z_j \right|^2} \\
&= \sqrt{\frac{1}{\tau^2} z_j^2 (1 - z_j)^2 + \sum_{i \neq j} \frac{1}{\tau^2} z_i^2 z_j^2} \\
&= \sqrt{\frac{1}{\tau^2} \left( z_j^2 (1 - z_j)^2 + \sum_{i \neq j} z_i^2 z_j^2 \right)} \\
&= \sqrt{\frac{1}{\tau^2} z_j^2 \left( (1 - z_j)^2 + \sum_{i \neq j} z_i^2 \right)} \\
&\leq \sqrt{\frac{1}{\tau^2} z_j^2 \left( (1 - z_j)^2 + 1 \right)} \\
&\leq \sqrt{\frac{1}{\tau^2} z_j^2 (1 + 1)} \\
&= \frac{\sqrt{2}}{\tau} z_j.
\end{aligned}
$$

### 7.4 Further Experiments

In the following, we illustrate further experiments. In particular, we created a new dataset for multi-hop reasoning and compared our model to its base model without discretization.

Table 6: The results of the path query task on the newly created multi-hop reasoning dataset based on FB15K-237. Our model that discretizes the score function of the base model performs much better on the path query task and generalizes perfectly to paths up to length 10.

| | MRR | | | | | MRR (etrapolation) | | | | |
|---|---|---|---|---|---|---|---|---|---|---|
| Model | 1 | 2 | 3 | 4 | 5 | 6 | 7 | 8 | 9 | 10 |
| ComplEx | **32.9** | 30.6 | 24.7 | 20.8 | 16.5 | 15.1 | 12.2 | 10.5 | 8.8 | 7.1 |
| ComplEx-C | 31.3 | 35.2 | 36.9 | 37.8 | 35.2 | 31.9 | 25.7 | 19.4 | 14.1 | 9.7 |
| Ours, $\tau = 4$ | 26.8 | **52.3** | **48.8** | **51.4** | **52.6** | **54.2** | **54.8** | **54.7** | **54.9** | **54.7** |

### 7.4.1   Multi-Hop Reasoning over FB15K-237

Since the benchmarks in Table 3 are based on Freebase and Wordnet, which have several issues, we also created a new dataset based on FB15K-237 [34] using the methodology of Guu et al. [10]. Particularly, we built the graph consisting of all training triples from the original dataset and sampled a starting entity uniformly. We then sampled an incident relation uniformly and sampled the next entity uniformly from all entities reachable via this relation. Continuing this way, we created a dataset of $272,115$ training paths of length (number of relations) $2, 3, 4, 5$, respectively. We repeated this procedure on the graph consisting of all triples (not only training triples) and removed all duplicates to create $17,535$ validation paths of length $2, 3, 4, 5$, and $20,466$ test paths of length $2, \ldots, 10$. Finally, we added all of the FB15K-237 triples to the dataset as paths of length $1$.

We use a slight variation of our model for Freebase and Wordnet to optimize the training in both directions of the path. As baselines, we take the same model without the discretization module and train it first on triples only (ComplEx) and then on all paths (ComplEx-C). We validate the models using filtered MRR (Bordes et al. [3]) on all validation paths and report test results on all paths for each length individually. Table 6 lists the results of the experiments on the new FB15K-237 based dataset. The models based on discrete-continuous components achieve improvements of up to $49\%$ compared to the baselines. Even more pronounced is the ability of these models to generalize: the accuracy does not drop even when tested on paths twice the length of those seen during training. The performance gap between $1$ relation and $2$ relations is mainly due to the fact that the test paths of length $1$ purely consist of triples from the test graph while all other paths consist of all triples from the full knowledge graph.

### 7.5   Error Bars for Multi-Hop Reasoning over Knowledge Graphs

In Table 7 we give results including the error bars for the experiments depicted in Table 3 and Table 4. The results of Table 3 are can be found in the row Paths $\leq 5$.

Table 7: Error bars for the experiments in Table 3 and Table 4.

| | | WordNet | | Freebase | |
|---|---|---|---|---|---|
| Paths | Model | MQ | H@10 | MQ | H@10 |
| $\leq 5$ | Ours | $94.4 \pm 0.0$ | $64.3 \pm 0.11$ | $89.6 \pm 0.46$ | $68.7 \pm 0.32$ |
| $\leq 4$ | Ours | $94.3 \pm 0.14$ | $64.7 \pm 0.11$ | $88.8 \pm 1.29$ | $69.4 \pm 0.72$ |
| $\leq 3$ | Ours | $93.4 \pm 0.04$ | $65.0 \pm 0.11$ | $88.3 \pm 1.85$ | $68.5 \pm 0.46$ |
| $\leq 2$ | Ours | $90.2 \pm 0.04$ | $62.0 \pm 0.04$ | $87.6 \pm 0.62$ | $68.6 \pm 0.36$ |
| $\leq 1$ | Ours | $81.1 \pm 0.22$ | $52.2 \pm 0.17$ | $82.1 \pm 0.54$ | $63.9 \pm 0.6$ |

### 7.6   Accuracy Curves of the Accuracies in Figure 4

To obtain the accuracies in Figure 4, we repeated each experiment $8$ times. For both of the experiment setups, TM and TauAnn, we found a single seed each, for which we noticed no training at all. Thus we repeated these experiments a ninth time. We depict the accuracy curves of all 9 runs of TM in Figure 7 and of TauAnn in Figure 8.

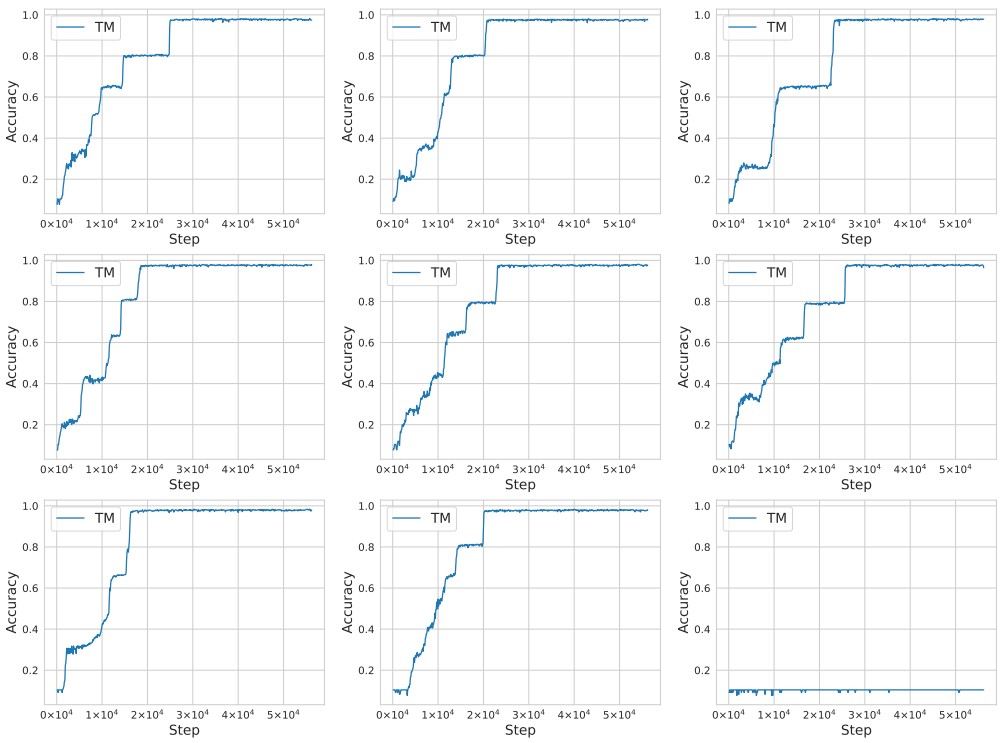

Figure 7: All 9 TM accuracies from the ablation study depicted in Figure 4.

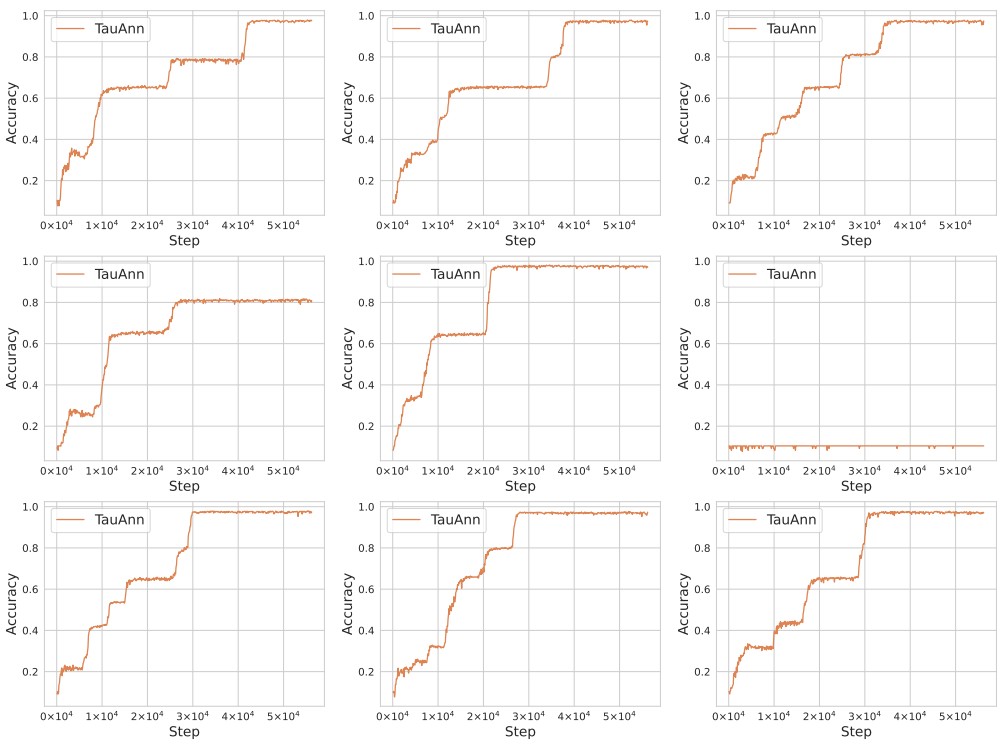

Figure 8: All 9 TauAnn accuracies from the ablation study depicted in Figure 4.