# OpenReview forum: "Efficient Learning of Discrete-Continuous Computation Graphs"
_NeurIPS.cc/2021/Conference — NeurIPS 2021 Poster_

### Official Review · Reviewer_UQHi · 2021-07-12

**Rating:** 5
**Confidence:** 4

**Summary:**

This paper tackles the learning of a computation graph with possibly multiple discrete and continuous nodes. The authors argue the challenge for effective learning is the gradient vanishing in Gumbel-softmax in discrete nodes, and propose two techniques, annealing parameters in Gumbel-Softmax and introducing residual dropout connections, to deal with the challenge. Empirical experiments show the effectiveness of the proposed method.

**Limitations And Societal Impact:**

The authors adequately addressed the limitations and potential negative societal impact.

**Main Review:**

This work addresses the task of learning in a computation graph with possibly multiple discrete and continuous nodes. Particularly, the authors consider discrete nodes instantiated as Gumbel-softmax, and argue that the vanishing gradient on Gumbel-softmax is the challenge posed to effective learning that should be addressed. This overall motivation is an important one, yet is it not un-explored. For example, the balance between approximation quality and saturation/vanishing gradients has already been noted [1] and compared [2], and improvements have been proposed [2]. These existing works are not present in the current draft, and the authors are encouraged to include them to give a better overview of the topic and the motivation behind it. The authors also argue for interpretability as an advantage for the proposed tasks throughout the draft (e.g. L4, L38, L253, Table 5), without elaboration. It may be better to demonstrate interpretability though either examples and/or visualization in one of the experiments to give readers a better understanding.

To tackle the challenge, the authors proposed two techniques, namely TM (Temperature Matching and Inverse Scale Annealing) and DR (Residual Dropout Connections), detailed in Section 2.1 and 2.2 respectively. From my understanding, TM is a newly proposed method --- actually the original Gumbel-Softmax paper [1] already adapts annealing in the training. Also, DR, or stochastic residual, has been proposed for vision tasks [3], while it's application to Gumbel-Softmax would be novel. Another question regarding DR would be, since only Gumbel-Softmax poses the issue of vanishing gradients, why not just skip-connecting the $\theta \to z$ part? Neitherness, the authors are encouraged to discuss these related works and compare the proposed method with them, as they are closely related to the claimed contributions.

The experiments are concluded thoroughly and the code is available (great job!) Probably due to the design of the methods, there are some hyperparameters to choose such as annealing rate (L484 "The temperatures are updated 10 times per epoch"), and schedule of decaying $\alpha$ that seem to be hard-coded in the code. The experiment section mentioned that hyperparameters are chosen following previous methods, but it may be more useful to conduct study on how they work to give readers more rationale behind these choices.


Overall, The work is clearly written and well organized andI think this is an interesting work Future works may draw inspirations (empirical methods that mitigate the gradient vanishing problem in Gumbel-Softmax) proposed in this manuscript. However, it still needs some improvements as detailed above, as a few discussions with existing works are missing.

References:

* [1] Jang et al. Categorical Reparameterization with Gumbel-Softmax. ICLR ;17 https://openreview.net/forum?id=rkE3y85ee
* [2] Shayer et al. Learning Discrete Weights Using the Local Reparameterization Trick. ICLR '18 https://openreview.net/forum?id=BySRH6CpW
* [3] Huang et al. Deep Networks with Stochastic Depth. ECCV '16. https://arxiv.org/pdf/1603.09382v2.pdf


**Time Spent Reviewing:**

3 hours

---

> ### Author Response · Authors · 2021-08-10
> **Answer to Reviewer UQHi**
>
> First of all, we want to thank you for the helpful and balanced review.
>
> > _For example, the balance between approximation quality and saturation/vanishing gradients has already been noted [Jang et al. [11]] and compared [1], and improvements have been proposed [1]. These existing works are not present in the current draft, and the authors are encouraged to include them to give a better overview of the topic and the motivation behind it._
>
> We thank the reviewer for these references and will adjust the related work session accordingly.
>
> > _The authors also argue for interpretability as an advantage for the proposed tasks throughout the draft (e.g. L4, L38, L253, Table 5), without elaboration. It may be better to demonstrate interpretability though either examples and/or visualization in one of the experiments to give readers a better understanding._
>
> Since we sample discrete intermediate latent representations during test time (using straight-through), we can read them out directly. See for example line 241 and the column “Inter. acc.” in Table 1 for the results on intermediate representations.
>
> > _From my understanding, TM is a newly proposed method --- actually the original Gumbel-Softmax paper [Jang et al. [11]] already adapts annealing in the training. Also, DR, or stochastic residual, has been proposed for vision tasks [2], while it's application to Gumbel-Softmax would be novel. […] Neitherness, the authors are encouraged to discuss these related works and compare the proposed method with them, as they are closely related to the claimed contributions._
>
> The original Gumbel-softmax paper adapts the standard annealing of the softmax temperature $\\tau$. In contrast, we discuss the advantages of keeping $\\tau$ constant and instead increasing the Gumbel scale $\\beta$ in relation to $\\tau$. Huang et al. [2] seems interesting related work. They use stochastic residuals in the opposite direction than we are, namely on the partial function instead of on the residual connections. This is a crucial difference since we do not utilize any residuals during test time. Nonetheless, we thank the reviewer for this reference and will add it to the related work session.
>
> > _Another question regarding DR would be, since only Gumbel-Softmax poses the issue of vanishing gradients, why not just skip-connecting the $\\phi\\rightarrow z$ part?_
>
> We do not fully understand the reviewer’s notion. Alternating between $\\phi$ as the parameter of the probability distribution and (with skip-connection) $\\phi$ as the sample of this distribution does not seem intuitive to us.
>
> > _The experiment section mentioned that hyperparameters are chosen following previous methods, but it may be more useful to conduct study on how they work to give readers more rationale behind these choices._
>
> We agree with the reviewer that describing some experimental setups in more detail might improve clarity. We will change this in the updated version of the paper.
>
> ---
>
> [1] Shayer, O., Levi, D. and Fetaya, E., 2017. Learning discrete weights using the local reparameterization trick. 6th International Conference on Learning Representations, ICLR 2018, Vancouver, BC, Canada, April 30 - May 3, 2018.
>
> [2] Huang, G., Sun, Y., Liu, Z., Sedra, D. and Weinberger, K.Q., 2016, October. Deep networks with stochastic depth. In European conference on computer vision (pp. 646-661). Springer, Cham.

---

### Official Review · Reviewer_DtA3 · 2021-07-16

**Rating:** 7
**Confidence:** 2

**Summary:**

Motivated by the vanishing gradient problem of multiple discrete operations that leverage the Gumbel Softmax trick for training, this paper proposes two new tricks.  First, the scale parameter of the Gumbel distribution is annealed across training rather than annealing the temperature parameter as is common in most prior works.  Second, residual connections are stochastically dropped via another annealing strategy so that gradients are non-vanishing in the beginning but eventually the residual connections are removed. Multiple empirical experiments demonstrate that the method is better at extrapolating than the baselines.


**Limitations And Societal Impact:**

- The paper seems adequate.

**Main Review:**

**Summary of review**
This paper introduces two novel tricks for complex discrete-continuous networks and shows strong results for extrapolation. While some writing could be improved and more explanation added, the paper seems to make useful contributions.

**Strengths:**
- The paper connects the interplay between the scale parameter of the Gumbel distribution and the temperature parameter of the softmax operation. In particular, it seems that annealing the Gumbel $\beta$ parameter may be better than annealing the temperature parameter $\tau$.

- The paper introduces a simple but interesting modification to residual connections that combines the idea of residual networks and dropout.

- The empirical results demonstrate superior performance for extrapolation on multiple experiments.

**Weaknesses:**
- It seems that the annealing schedules for both proposed tricks is quite important. Could more discussion be given to the selection of annealing schedules?

- The problem with overfitting that motivates DropRes is not clearly explained.  It seems to merely be an empirical observation.  This should be explained more carefully and any intiution or explanation should be provided.

- Related to above, how does the Gumbel scale parameter directly affect things?  It doesn't seem to be connected theoretically in say equations 3-5.  Is this just an empirical observation and basic intuition that they are related?  Is there a more fundamental or mathematical relationship between these parameters?

**Other comments or questions**
- How do these two tricks work for a single discrete operator?  The paper seems to emphasize multiple times that a sequence of discrete operations is the key problem being tackled in the paper.  Thus, I'm wondering how much it makes a difference for the simpler cases.


**Time Spent Reviewing:**

1.25

---

> ### Author Response · Authors · 2021-08-10
> **Answer to Reviewer DtA3**
>
> We want to thank the reviewer for the motivating review.
>
> > _It seems that the annealing schedules for both proposed tricks is quite important. Could more discussion be given to the selection of annealing schedules?_
>
> We increase $\\alpha$ linearly with $\\alpha_0=0, \\alpha_t=\\min(1, ct)$, for some $c>0$. We increase beta by using the schedule $\\beta_t = \\tau(1- e^{-t\\gamma})$ for some $\\gamma>0$. The exact parameters are detailed in the appendix (for example lines 484-485). We agree with the reviewer’s comments that the annealing schedules are of importance and belong in the main paper.
>
> > _The problem with overfitting that motivates DropRes is not clearly explained. It seems to merely be an empirical observation. This should be explained more carefully and any intiution or explanation should be provided._
>
> As explained in lines 140+141, we remove the residual connections during test time. Thus, we sample purely discrete during test time. While the residual connections help with the vanishing gradients, we need to reduce the residuals to zero over time to close the gap between training and test time.
>
> > _Related to above, how does the Gumbel scale parameter directly affect things? It doesn't seem to be connected theoretically in say equations 3-5. Is this just an empirical observation and basic intuition that they are related? Is there a more fundamental or mathematical relationship between these parameters?_
>
> Intuitively, we want to make the model more frequently draw (relaxed) discrete samples that are low probability but which then incur a higher downstream loss and, therefore, allow the model to continue to learn. This is also demonstrated by the example plotted in Figure 3: while the probability has saturated, the model with higher scale parameters beta obtains more pronounced gradient signals precisely because more often than for a smaller value for beta, a (relaxed) configuration is sampled that incurs a downstream loss.
>
> > _How do these two tricks work for a single discrete operator? The paper seems to emphasize multiple times that a sequence of discrete operations is the key problem being tackled in the paper. Thus, I'm wondering how much it makes a difference for the simpler cases._
>
> We actually have tried our methods for computation graphs with single discrete nodes particularly for the Discrete VAE setting. We did not see improvements in this case what we believe is the expected outcome. Sigmoid activation functions also tend to cause vanishing gradient problems mainly in the setup of deep neural networks in which multiple sigmoid activations are used consecutively. Similarly, our methods have the most impact in stochastic computation graphs with multiple consecutive discrete nodes.

---

### Official Review · Reviewer_Thd9 · 2021-07-17

**Rating:** 5
**Confidence:** 4

**Summary:**

The authors consider the problem of learning in models that combine discrete and continuous operations through the use of stochastic sampling of categorical variables. In the case where the learning makes use of the Gumbel-Softmax reparametrization of the categorical distributions, the authors propose two modifications to the architecture of the network in order to improve training: 1) tempering the categorical distributions towards more uniformity in the later stages of training, and 2) including a stochastic pass-through (residual connection). The authors demonstrate the benefits of the proposed modifications on a couple of common tasks in the domain.

**Limitations And Societal Impact:**

Please see main review for limitations.

**Main Review:**


The authors propose techniques which are somewhat unsurprising, but they demonstrate good results on proposed applications. Unfortunately, the authors do not provide direct comparisons between baselines and their suggested techniques, making it difficult to identify the source of the improvement. This paper could be greatly improved by more directly focusing on the problem of gradient propagation through stochastic neurons, and in particular it would be helpful for the authors to directly compare with commonly used methods (e.g. see [1]). Given the lack of direct evidence for the effectiveness of the proposed methods, I do not believe that this paper is quite ready. Please see more detailed comments below.

On the empirical results. Given the main claims in the abstract (“we show that it is challenging to optimize [...], mainly due to vanishing gradients. We then propose two new strategies to overcome these challenges [...]”), I believe that the main contribution of this paper is to tackle the fairly general problem of gradient propagation through stochastic neurons. In this light, the empirical section of the paper should be reworked to better support these claims, with more emphasis on ablation and comparative studies (e.g. as presented in table 2, but for a large range of models and problems), rather than introducing the author’s own variations for standard problems, where the source of the improvement is not obviously identified. Additionally, such comparisons should attempt to include other commonly used strategies to propagating gradients through stochastic neurons: in particular, the author’s proposal for DropRes has some parallels to the straight-through estimators. Similarly, figure 4 should be extended for different combinations of models, problems and gradient estimators.

On the bias of the proposed methods. Compared to estimators of the gradient based on the score function, the reparameterized gradients for discrete stochastic neurons are slightly biased. The proposed methods in this paper further amplify this bias, by 1) increasing $\beta$ (effectively scaling down $\theta$ and $\tau$), and 2) considering an uncorrected residual connection. Given the presence of this bias, it would be interesting to compare these methods to standard methods (e.g. based on the score-function estimator a.k.a. REINFORCE and their variance-reduced counterparts) to better understand the impact of these methods, and e.g. whether the improvements are due to better gradient propagation and training behavior, or some implicit regularization due to the introduced bias.


[1] Estimating or propagating gradients through stochastic neurons for conditional computation. Bengio, Leonard and Courville. ArXiv 2013

Edit after author response

I thank the authors for their response. I believe that ablation studies should be the primary empirical justification for claims advanced in this paper, and I strongly encourage the authors to consider expanding the ablation studies and reducing the number of comparison with custom-tuned models, which although interesting, do not necessarily provide evidence for the current claims as they do not clearly demonstrate the source of the improvement.


**Time Spent Reviewing:**

6

---

> ### Author Response · Authors · 2021-08-10
> **Answer to Reviewer Thd9**
>
> We would like to thank the reviewer for the balanced review.
>
> > _Unfortunately, the authors do not provide direct comparisons between baselines and their suggested techniques, making it difficult to identify the source of the improvement. This paper could be greatly improved by more directly focusing on the problem of gradient propagation through stochastic neurons, and in particular it would be helpful for the authors to directly compare with commonly used methods (e.g. see [1])_
>
> While [1] is important related work that we will incorporate in the main paper, it focuses on execution paths with single stochastic nodes. The key difference is that we focus on efficiently training models with multiple consecutive stochastic nodes on the execution paths.
>
> > _In this light, the empirical section of the paper should be reworked to better support these claims, with more emphasis on ablation and comparative studies (e.g. as presented in table 2, but for a large range of models and problems), rather than introducing the author’s own variations for standard problems, where the source of the improvement is not obviously identified._
>
> We agree with the reviewer’s comment as well as with the other reviewers that such an ablation study would provide further insights. We will include additional ablation experiments in the updated version of the paper.
>
> > _The proposed methods in this paper further amplify this bias, by 1) increasing $\\beta$ (effectively scaling down $\\phi$ and $\\tau$), and 2) considering an uncorrected residual connection. Given the presence of this bias, it would be interesting to compare these methods to standard methods (e.g. based on the score-function estimator a.k.a. REINFORCE and their variance-reduced counterparts) to better understand the impact of these methods, and e.g. whether the improvements are due to better gradient propagation and training behavior, or some implicit regularization due to the introduced bias._
>
> We thank the reviewer for this comment and think that an analytical comparison between our biased model and related unbiased methods can be interesting future work.
>
> ---
> [1] Bengio, Y., Léonard, N., & Courville, A. (2013). Estimating or propagating gradients through stochastic neurons for conditional computation. arXiv preprint arXiv:1308.3432.

---

### Official Review · Reviewer_Uy2p · 2021-07-18

**Rating:** 4
**Confidence:** 4

**Summary:**

Summary.

- This paper introduces two techniques to help with learning neural network architectures that embed two or more layers of categorical probability distributions (parameters of those distributions are learnt!).
- The first method is Temp-Match: The suggestion is to separately tune the variance of the Gumbel noise and the temperature parameter.
- The second method are dropout-residual connections. The authors add residual connections to connect the input to the output stochastically for a given discrete-continuous component (Figure 2).



**Main Review:**

**Strengths**
- Clarity. The paper is well-written overall and I like the structure.
- Significance. Studying gradient estimation for discrete-continuous computation graphs is interesting.
- Originality. I am not an expert on dropout and residual connections, but I have not seen dropout-residual connections as presented in this paper before.

**Weaknesses**
- Originality. I feel the relationship between the temperature parameter and the Gumbel noise variance is known widely, even though I don’t have a concrete reference on hand right now.
- Quality. I do not find the analysis of the vanishing gradient problem when using Gumbel-Softmax convincing.
- Quality, Significance. From the argument and evidence presented, I wouldn’t use Temp-Match.
- Quality. I believe the experiments need to be revised to make a stronger case for the methods presented (e.g. more complex models with more stochastic layers, see below).

**Decision**
- Ok, but not good enough.
As is, unfortunately, I do not think the paper is ready for acceptance.
I would like to make some suggestions for improvement below.

**Suggestions**

*As is, I do not find the analysis of the vanishing gradient problem when using Gumbel-Softmax convincing.*
- The authors say that gradients may vanish whenever the probability saturates.
But when the probability saturates, the categorical distribution becomes degenerate, the stochasticity vanishes, the output is deterministic.
So my question: Isn’t this a setting in which I may not want to model a latent categorical distribution in the first place, and could use a deterministic function instead?
My issue with the toy example presented in Figure 3 is similar.
This toy example is presenting an edge case, where the ideal distribution is degenerate, since target is $(1, 0)$.
Why not present the same case as the REBAR paper, which seems very similar in set-up but didn’t pick an edge case. To what extent is the problem present there?

*From the argument and evidence presented, I probably wouldn’t use Temp-Match.*
- I am missing a discussion on the theoretical side, that highlights that for every $(\tau, \beta)$ pair, there is a pair $(\alpha, \tau’)$ with $\beta = 1$ that gives rise to the same Gumbel-Softmax distribution as $(\tau, \beta)$. Here, $\alpha$ is a parameter that scales the logits. The reason for this is basically, that $\tau$ inversely scales both logits and Gumbels. To decouple scaling both logits and Gumbels together, one can introduce either an additional parameter to rescale the Gumbels (this is $\beta$) or an additional parameter to rescale the logits (I used $\alpha$ for that above). In practice, since the logits are outputs from a neural network with learnable parameters, the $\alpha$ is implicit and is learnt. My questions: Why is Gumbel scaling better than logit scaling? Why should $\alpha$ (or $\beta$) not be learnt?
- Further, Figure 3 seems to present some evidence for the benefits of tuning $\beta$. But $\tau$ is held fixed. Isn’t the argument that is required compare “tuning $\tau$” vs “tuning $\tau$ and $\beta$” (both with a schedule ideally). There should also be a loss curve, to make the case that “\tuning $\tau$ and $\beta$” vs “tuning $\tau$” yields significant improvements that justify an extra hyperparameter.
-Finally, Temp-Match is evaluated in an ablation study. I like that. But the standard deviations are so high, that I do not see any evidence for benefits from TempMatch. Or am I misinterpreting Table 2?

*The focus of this paper is on complex networks with multiple discrete components, but the experimental section is underdelivering.*
- To make a stronger case for the setting and methods of the paper, I would like to see settings and models that are more ambitious and practical. As a concrete suggestion, I would recommend looking into the model Gumbel-Tree-LSTM from Choi et al. (2017). This model is much better tailored to the ListOps dataset than the GNN model considered by the authors.
It also requires much more stochastic decisions, namely #number of tokens - 1, rather than a fixed set, so I feel is more relevant experimentally to the methods presented. Other papers that cite ListOps also include this model or variants that I believe are generally viewed as more suitable for unsupervised parsing on ListOps. I do not understand why the authors considered a GNN architecture instead and modified the ListOps dataset to make the GNN practical.

**Minors.**
- Gumbel hard setting -> I think generally this is referred to as Gumbel straight-through. See for example, Jang et al. (2016) or Paulus et al. (2021).

**References.**
- Jang et al. (2016). Categorical reparameterizatoin with Gumbel-Softmax.
- Choi et al. (2017). Learning to compose task-specific tree structures.
- Paulus et al. (2021). Rao-Blackwellizing the Straight-Through Gumbel-Softmax estimator.


EDIT.
I read through the response of the authors, but I am keeping my score unchanged.
Unfortunately, I am of the opinion that more work is required to improve the paper before it is ready for submission along the lines I suggested above.


**Time Spent Reviewing:**

I don't stop a clock, but probably 4 hours.

---

> ### Author Response · Authors · 2021-08-10
> **Answer to Reviewer Uy2p**
>
> We would like to thank the reviewer for the helpful review.
>
> > _The authors say that gradients may vanish whenever the probability saturates. But when the probability saturates, the categorical distribution becomes degenerate, the stochasticity vanishes, the output is deterministic. So my question: Isn’t this a setting in which I may not want to model a latent categorical distribution in the first place, and could use a deterministic function instead?_
>
> As discussed in lines 37-39 and shown as one of the main aims in the experiments, the categorical distribution yields more advantages than “stochasticity”. Using discrete nodes, we obtain latent representations that exhibit improved generalization behavior and are more interpretable.
>
> > _My issue with the toy example presented in Figure 3 is similar. This toy example is presenting an edge case, where the ideal distribution is degenerate, since target is $(0, 1)$ . Why not present the same case as the REBAR paper, which seems very similar in set-up but didn’t pick an edge case. To what extent is the problem present there?_
>
> In Figure 3, we give a simple intuition for the influence of the Gumbel scale $\\beta$ on the Gumbel-softmax distribution. Thus, we chose a straightforward toy example. Choosing a set-up that is known to be generally hard for the Gumbel-softmax distribution, as the reviewer proposes, does not seem intuitive for this experiment.
>
> > _I am missing a discussion on the theoretical side, that highlights that for every $(\\tau,\\beta)$ pair, there is a pair $(\\alpha,\\tau’)$ with $\\beta=1$  that gives rise to the same Gumbel-Softmax distribution as . Here, $\\alpha$ is a parameter that scales the logits. The reason for this is basically, that $\\tau$ inversely scales both logits and Gumbels. To decouple scaling both logits and Gumbels together, one can introduce either an additional parameter to rescale the Gumbels (this is $\\beta$) or an additional parameter to rescale the logits (I used $\\alpha$ for that above). In practice, since the logits are outputs from a neural network with learnable parameters, the $\\alpha$ is implicit and is learnt. My questions: Why is Gumbel scaling better than logit scaling? Why should $\\alpha$ (or $\\beta$) not be learnt?_
>
> We use the parameters $\\tau$ and $\\beta$ in the way we do because we think it makes the behavior of increasing $\\beta$ (while keeping $\\tau$ fixed) easier to understand. Intuitively, we want to make the model more frequently draw (relaxed) discrete samples that are low probability but which then incur a higher downstream loss and, therefore, allow the model to continue to learn. See also our discussion about this in lines 110-114. As the reviewer correctly deduced, one could equivalently scale the two parameters $\\alpha$ and $\\tau$ simultaneously while keeping $\\beta$ constant. We prefer scaling the single parameter $\\beta$ that we understand and keeping $\\alpha$ constant, which is also common practice in for example transformers with $\\alpha = \\sqrt{d}$.
>
> > _Further, Figure 3 seems to present some evidence for the benefits of tuning $\\beta$. But $\\tau$ is held fixed. Isn’t the argument that is required compare “tuning $\\tau$” vs “tuning $\\tau$ and $\\beta$” (both with a schedule ideally). There should also be a loss curve, to make the case that “\tuning $\\tau$ and $\\beta$” vs “tuning $\\tau$” yields significant improvements that justify an extra hyperparameter._
>
> We agree with the reviewer’s comment that such an ablation study would provide further insights. We will include additional ablation experiments for additional tasks in the updated version of the paper.
>
> > _Finally, Temp-Match is evaluated in an ablation study. I like that. But the standard deviations are so high, that I do not see any evidence for benefits from TempMatch. Or am I misinterpreting Table 2?_
>
> The main ablation study of TempMatch can be seen in the last two lines of Table 2 (without the use of DropRes). Using TempMatch significantly improves over a model that uses neither of the proposed methods.
>
> > _As a concrete suggestion, I would recommend looking into the model Gumbel-Tree-LSTM from Choi et al. (2017). This model is much better tailored to the ListOps dataset than the GNN model considered by the authors._
>
> Choi et al. [1] seems to be interesting related work. We will have a look into it. For our ListOps experiments, we followed Paulus et al. [20], which is most related to our work.
>
> > _Gumbel hard setting -> I think generally this is referred to as Gumbel straight-through. See for example, Jang et al. (2016) or Paulus et al. (2021)._
>
> Thank you. We will change this accordingly in our paper.
>
> ---
>
> [1] Choi, J., Yoo, K.M. and Lee, S.G., 2018, April. Learning to compose task-specific tree structures. In Thirty-Second AAAI Conference on Artificial Intelligence.

---

### Decision · Program_Chairs · 2021-09-28

**Decision:**

Accept (Poster)

**Comment:**

This paper proposed two ways to optimize the computation graph that involves with multiple sequential discrete components, namely the dropout residual connection and the tuning of Gumbel noise. While the reviewers generally agree that the problem studied is an important one, there are several common concerns.
- The technical part of the paper, especially its novelty, the motivation and the rigorous analysis behind it.
- More convincing experiments, with better positioning among existing works and better design of the experiments.

After the authors' response, all the reviewers participated in the committee discussion, and the recommended decision was made based on the consensus of all the reviewers and AC. Despite that the paper cannot be accepted in the current form, the reviewers are optimistic about the paper and encourage the authors to further improve the paper based on reviewers' comments, especially on the motivation/analysis of the proposed method and the empirical evaluations.

**Consistency Experiment:**

NeurIPS has a long history of experimentation. In 2014, NeurIPS ran an experiment in which 10% of submissions were reviewed by two independent committees to quantify the randomness in the review process. This year, we repeated a variant of this experiment to see how the quality of the review process has changed over time.  This paper was part of the experiment and was therefore assigned to two committees (consisting of reviewers, an Area Chair, and a Senior Area Chair) that reached independent decisions.  If both committees made the same recommendation, this recommendation was followed. If a single committee recommended acceptance, the paper was accepted (with the exception of a few cases in which the other committee identified what we considered a fatal flaw, e.g., an error in a key result).

This copy’s committee reached the following decision: **Reject**

The other committee assigned to the paper recommended **Accept (Poster)**.  You can find the other set of reviews, along with any follow up discussion with the authors here:
https://openreview.net/forum?id=TLIHuw3gcQB